# PAT3D: P̲HYSICS-A̲UGMENTED T̲EXT-TO-3D SCENE GENERATION

**Guying Lin**[1]   **Kemeng Huang**[2,1]   **Michael Liu**[1]   **Ruihan Gao**[1]   **Hanke Chen**[1]   **Lyuhao Chen**[1]
**Beijia Lu**[1]   **Taku Komura**[2]   **Yuan Liu**[3]   **Jun-Yan Zhu**[1]   **Minchen Li**[1,4]

[1]Carnegie Mellon University   [2]The University of Hong Kong
[3]The Hong Kong University of Science and Technology   [4]Genesis AI

{guyingl, mliu6, ruihang, lyuhaoc, beijialu}@andrew.cmu.edu,
kmhuang@connect.hku.hk, {hankec, junyanz}@cs.cmu.edu, taku@cs.hku.hk,
yuanly@ust.hk, minchernl@gmail.com

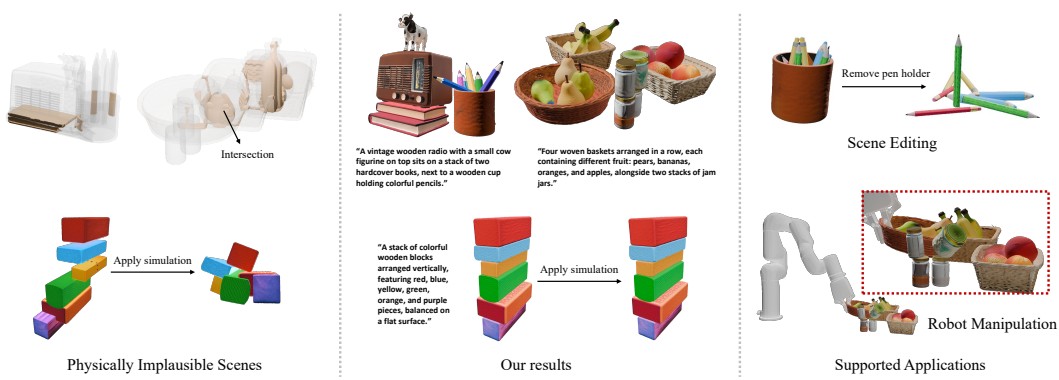

Figure 1: PAT3D is the first text-to-3D scene generation framework that produces **simulation-ready** and **intersection-free** results. The left column shows results from direct depth-based arrangements, which suffer from object interpenetrations (top) and collapse under simulation due to inconsistent layouts (bottom). The middle column presents PAT3D results, where physically valid layouts remain stable under simulation. These high-quality scenes are immediately usable for downstream applications, including scene editing and robotic manipulation (right).

## ABSTRACT

We introduce PAT3D, the first physics-augmented text-to-3D scene generation framework that integrates vision–language models with physics-based simulation to produce physically plausible, simulation-ready, and intersection-free 3D scenes. Given a text prompt, PAT3D generates 3D objects, infers their spatial relations, and organizes them into a hierarchical scene tree, which is then converted into initial conditions for simulation. A differentiable rigid-body simulator ensures realistic object interactions under gravity, driving the scene toward static equilibrium without interpenetrations. To further enhance scene quality, we introduce a simulation-in-the-loop optimization procedure that guarantees physical stability and non-intersection, while improving semantic consistency with the input prompt. Experiments demonstrate that PAT3D substantially outperforms prior approaches in physical plausibility, semantic consistency, and visual quality. Beyond high-quality generation, PAT3D uniquely enables simulation-ready 3D scenes for downstream tasks such as scene editing and robotic manipulation.

# 1 INTRODUCTION

The ability to generate realistic and editable 3D scenes from natural language has broad applications across a variety of domain including virtual reality, robotics, digital twins, and content creation. Recent advances in diffusion and autoregressive generative models have significantly pushed the boundaries of text-to-3D scene generation, making it possible to synthesize high-quality object geometry and compelling visual content Lin et al. (2023); Metzer et al. (2023); Michel et al. (2022); Poole et al. (2023); Chen et al. (2025b; 2024a); Huang et al. (2024); Gao et al. (2024). However, despite these advances, existing approaches struggle to ensure that generated scenes exhibit *physical plausibility* – a critical requirement for downstream applications that demand interaction, simulation, or building a real-world correspondence.

In particular, current 3D scene generation pipelines Huang et al. (2024); Gao et al. (2024) often treat layout composition as a purely geometric problem, omitting physical reasoning entirely or using simple heuristics to prevent unfavored physical interaction such as object intersection. Due to the lack of explicit constraints from physics, this leads to common issues such as floating, unstable stacking, and incorrect support relations, ultimately limiting scene realism and usability. Earlier efforts have incorporated physical constraints to enhance single-object stability Guo et al. (2024); Chen et al. (2024c), or used video diffusion priors for plausible dynamics Zhang et al. (2024), but none of these methods address the complex spatial dependencies and contact interactions required for stable and semantically coherent multi-object scenes.

One promising direction is to integrate physics-based simulation into the scene generation process to enhance physical realism. However, this approach introduces several challenges. First, objects must be represented as individually segmented 3D meshes to enable simulation of interactions under gravity and contact forces. Applying simulation to scenes represented by a single connected mesh is ineffective, as it fails to capture interactions between objects. Second, physics-based simulation requires a well-posed initial configuration, typically free of intersections, to avoid numerical instability and unrealistic behavior Li et al. (2020). Yet, identifying such an intersection-free starting state is nontrivial. Finally, even if the simulated scene is physically plausible, it may diverge from the intended semantics described in the input text, due to the multiplicity of valid static equilibria.

To address these challenges, we propose **PAT3D**, a *physics-augmented text-to-3D scene generation* framework that integrates differentiable rigid-body contact simulation into the generation pipeline. Given a text prompt, we first synthesize a reference image to reflect the spatial relations among objects. Individual objects are then generated and coarsely positioned using vision foundation models Bochkovskii et al. (2025); Kirillov et al. (2023b); Hunyuan3D (2025). Next, a vision-language model (VLM) Hurst et al. (2024) extracts the physical dependencies between objects from the reference image, which are then organized into a scene tree. PAT3D then produces an intersection-free initial configuration from the coarsely positioned 3D scene and scene tree through physics-guided refinement. This initialization deliberately introduces small gaps along the gravity direction for objects with parent–child relations in the scene hierarchy, simplifying intersection avoidance while preserving inferred spatial relations. These gaps are later resolved through simulation, allowing objects to settle naturally under gravity and contact forces while making slight, physically plausible adjustments to their spatial relations. Finally, differentiable simulation is applied to further optimize the layout, improving semantic consistency in the resulting scene.

We validate our method on diverse, contact-rich scenes and demonstrate its effectiveness against existing state-of-the-art 3D scene generation approaches through both qualitative and quantitative evaluations under visual quality and physical plausibility metrics. We further demonstrate that our generated scenes are readily editable and interactable through simulation, enabling physically plausible scene editing and direct construction of simulation environments for policy evaluation in robotic manipulation tasks. Our code and data are available at https://github.com/Simulation-Intelligence/PAT3D.

In summary, our main contributions include:

- We introduce **PAT3D**, the first physics-augmented text-to-3D scene generation framework that integrates vision–language models with physics-based simulation, achieving state-of-the-art physical plausibility, semantic consistency, and visual quality.

- We propose a physics-aware scene initialization module to prepare scenes for simulation. This module infers physical dependencies among objects, organizes them into a hierarchical scene tree, and converts the scene tree into intersection-free initial conditions for simulation.

- We develop a layout optimization strategy based on artificially time-stepped differentiable simulation, enabling efficient evaluation and differentiation of static equilibrium w.r.t initial layout.

## 2  RELATED WORK

**Single Object Generation.**  Building on the success of text-to-image generation models Rombach et al. (2022); Ramesh et al. (2022); Kang et al. (2023); Yu et al. (2022), there has been rapid progress in 3D generative models conditioned on text or images. A prominent class of methods leverages 2D diffusion priors for 3D generation Poole et al. (2023); Wang et al. (2023); Lin et al. (2023); Chen et al. (2023); Metzer et al. (2023); Wang et al. (2024); Sun et al. (2024); Long et al. (2024); Michel et al. (2022), with DreamFusion Poole et al. (2023) introducing Score Distillation Sampling (SDS) to optimize 3D representations using gradients from 2D diffusion models. Subsequent works have extended SDS with multi-view diffusion models, improving both 3D generation quality and single-view reconstruction Liu et al. (2023a); Wang & Shi (2023); Shi et al. (2024); Liu et al. (2024b); Zhou & Tulsiani (2023); Liu et al. (2024a); Long et al. (2024); Shi et al. (2023); Liu et al. (2024d). Another research direction trains large-scale transformers to generate 3D shapes in a feed-forward manner Hong et al. (2024); Li et al. (2024); Xu et al. (2024b); Tochilkin et al. (2024), relying on curated, large-scale 3D asset datasets. While these models can generate visually compelling shapes, they often ignore the physical properties, such as stability, of the object, which are essential for real-world applications. To address this, recent efforts have incorporated physics-based simulation into the generation pipeline to produce self-supporting 3D objects by optimizing physical attributes such as mass distribution Guo et al. (2024); Chen et al. (2024c); Yan et al. (2024); Cai et al. (2024). Additionally, PhysDreamer Zhang et al. (2024), optimizes physical properties like Young's modulus and initial velocity to generate dynamic motions that are both visually plausible and physically grounded, guided by video diffusion priors.

**Scene Generation.**  While single-object generation methods produce visually appealing assets, they often lack scale awareness and spatial grounding, making scene composition challenging. The primary bottlenecks in 3D scene generation include decomposing scenes into individual assets, estimating their relative scale and pose, and ensuring physical feasibility (e.g., contact, stability). Several works address these challenges through multi-stage pipelines. Early methods such as Vilesov et al. (2023); Chen et al. (2024b); Han et al. (2024) adopt object-centric reconstruction followed by layout and geometry optimization using physical constraints or differentiable rendering. Shriram et al. (2024) lifts the scene image to 3D point clouds as a whole, inpaints occluded regions, and refines appearance using 2D diffusion priors. Recently, Large Language Models (LLMs) and VLMs are increasingly leveraged to infer spatial relations and scene structure. Gao et al. (2024) constructs a scene graph with objects as nodes and their relations as edges. Zhou et al. (2024) uses a VLM to generate a coarse layout, which is subsequently refined with rendering losses and physical constraints. Yao et al. (2025) infers a scene graph describing simplified pairwise relationships between objects, and use them to optimize object poses and scales. Wang et al. (2025) similarly leverages LLM-based reasoning to query objects' relative sizes and physical properties, enabling more plausible scene layouts. However, none of these methods can ensure physically accurate contact handling or maintain physical stability in the generated scene. Scene-level optimization under SDS loss is commonly used for joint geometry-text alignment Zhou et al. (2025b), while Huang et al. (2024) and Xu et al. (2024a) explore multi-instance and 4D compositional generation guided by spatial and trajectory priors. Building on similar SDS-based refinement, Zhou et al. (2025a) generates compositional 3D scenes from 2D layouts, enabling controllable instance placement but still lacking accurate modeling of physical interactions between objects. Other approaches incorporate physical property estimation into 3D representations, either from visual cues Zhao et al. (2024) or explicit user input Chen et al. (2025a), to support dynamic simulation or interaction Liu et al. (2024c). Recent systems such as Blender-MCP, Huang et al. (2025b), and Li et al. (2025) integrate LLM reasoning into graphics tools and simulation engines, enabling interactive behavior and fine-grained control. Sun et al. (2025) propose LayoutVLM, which uses a VLMs to generate differentiable spatial rela-

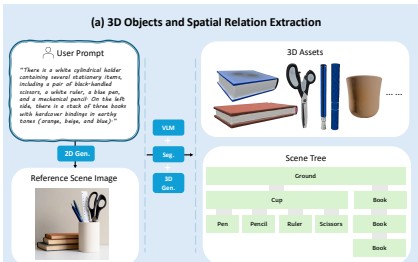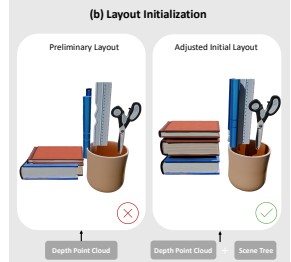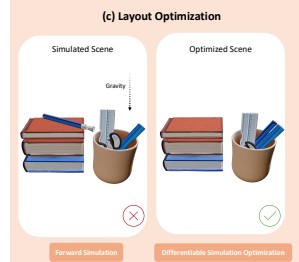

Figure 2: **Overview of our text-to-3D scene generation pipeline.** (a) Given an input text, a reference image is first generated to capture spatial relations among objects, from which 3D assets are generated using vision foundation models, and a scene tree is extracted using a VLM. (b) Assets are arranged into an initial layout using 3D priors from monocular depth estimation (left), then refined with the scene tree to produce an intersection-free configuration for simulation (right). (c) Forward simulation ensures physical plausibility but may distort semantics (left). We address this with simulation-in-the-loop optimization, enforcing semantic consistency and physical validity (right).

tions and jointly optimize 3D layouts in indoor scenes. However, most existing scene generation methods focus primarily on layout composition. They either omit physical reasoning altogether or incorporate only simple physics priors to prevent object interpenetration, without modeling accurate contact interactions or ensuring physically stable scene layouts. We thus address this gap by novelly augmenting text-to-3D scene generation with differentiable rigid body contact simulation.

## 3 METHOD

Our framework comprises three stages: *3D object and spatial relation extraction* (subsection 3.1), where 3D assets are generated from text and its spatial relation are organized into a scene tree; *layout initialization* (subsection 3.2), which first arranges generated assets using monocular depth priors obtained from refernece image and uses scene tree to refine them into an intersection-free configuration; and *layout optimization* (subsection 3.3), where a simulation-in-the-loop optimization procedure is applied to ensure physical plausibility and improve semantic consistency of 3D scene.

### 3.1 3D OBJECT AND SPATIAL RELATION EXTRACTION

Since directly producing both 3D objects and layouts with text-to-3D models and LLMs often fails to capture complex spatial relations, we instead employ a text-to-image model to generate a reference image that guides object generation and scene tree construction. See Figure 2(a).

#### 3.1.1 3D OBJECTS GENERATION

To generate individual objects for the scene specified by the text prompt, a VLM is queried with the reference image to obtain object class labels, and the image is segmented with Grounded-SAM Kirillov et al. (2023a); Ren et al. (2024); Liu et al. (2023b) accordingly. Based on the segmented object regions, we further prompt the VLM to generate detailed text descriptions encompassing object semantics, material, color, and orientation. These descriptions are fed into a text-to-3D pipeline Hunyuan3D (2025) to synthesize high-quality, textured 3D assets that are both semantically consistent and visually realistic.

#### 3.1.2 SPATIAL RELATION EXTRACTION

We then extract the relative spatial relations among objects in the scene from the reference image and analyze their physical dependencies. This information provides essential guidance for intersection-free layout initialization (subsection 3.2) and subsequent optimization (subsection 3.3). Specifically, for each pair of segmented objects that appear with similar horizontal positions and adjacent vertical positions in the reference image, we prompt a VLM to infer their dependency along the gravity axis, identifying relations such as on, contain, and support. The resulting pairwise relations are then

organized into a hierarchical scene tree that encodes how objects support one another under gravity. Starting with the ground as the root node, we traverse the scene and iteratively add objects as nodes in the tree. For each unvisited object that has a direct physical dependency with an existing node, we insert it as a child of that node. This recursive process continues until all objects have been included. Additional details are provided in Algorithm 1, and an example is shown in Figure 2(a).

## 3.2 Layout Initialization

To obtain an intersection-free and semantically consistent initial layout for the subsequent simulation-in-the-loop optimization, we first compute the translational and scaling transformations to build a preliminary layout consistent with the reference image, and then refine it using the extracted scene tree to ensure no object intersection and stronger physical constraints. See Figure 2(b).

### 3.2.1 Preliminary Layout

Our straightforward approach to arranging the objects generated in subsubsection 3.1.1 into a layout consistent with the reference image is to back-project the 2D reference image with depth estimation to obtain each object's 3D point cloud. Scaling and translational transformations can then be computed by aligning the object's center with the centroid of its point cloud. In practice, however, heavy occlusions in the 2D image make scaling unreliable when derived directly from partial point clouds. To address this, we first query the VLM to identify the least occluded object in the scene and use it as an anchor to compute a global scaling transformation for the entire scene. We then compute relative scaling for the other objects by prompting the VLM to inpaint occluded regions of the 2D image and estimating scaling factors from the bounding boxes of the inpainted objects. Each object's final transformation is obtained by combining the global and relative scaling factors, followed by alignment with the projected 3D point cloud. This procedure produces the preliminary layout shown in Figure 2(b).

### 3.2.2 Refined Initial Layout

To refine the layout under physical dependency constraints and ensure non-intersection, we traverse the scene tree in a breadth-first manner and apply horizontal and vertical refinements at each node.

**Horizontal refinement.** We enforce two rules: (1) Parent–child: the projection of the child must lie entirely within that of the parent (e.g., fruits inside a basket); (2) Sibling: objects sharing the same parent must have non-overlapping projections (e.g., a vase, plate, and fork on a table).

**Vertical refinement.** Each child is lifted above the bounding box of its parent along the gravity axis, preventing intersections.

This simple strategy, compared with more complex optimization methods, efficiently resolves intersections while preserving semantic constraints, providing favorable initial conditions for simulation. The refined results are shown in Figure 2(b).

## 3.3 Layout Optimization

After simulation, gravity causes child objects to fall onto or into their respective parents, and sibling objects naturally adopt physically plausible poses. However, due to complex inter-object interactions, simulation alone may cause the scene to deviate from its intended semantics. To address this, we introduce a simulation-in-the-loop optimization to improve semantic consistency in the simulated scene. See Figure 2(c).

Specifically, we refine our intersection-free initialization $q_0$ so that the final equilibrium state $q_{n+1}$ better aligns with the scene tree:

$$\min_{q_0} L(q_{n+1}(q_0)) \quad \text{s.t.} \quad f(q_{n+1}) = 0, \tag{1}$$

where $L$ measures semantic inconsistency and $f$ denotes the net force on all objects..

For each object $i$ with container $t$, we define its projected bounding box on the horizontal plane as $\text{BBox}_i = \{\mathbf{p}^i_{\min}, \mathbf{p}^i_{\max}\}$. The local loss penalizes deviations of the corners of $i$ from $\text{BBox}_t$:

$$l_i = d(\mathbf{p}^i_{\min}, \text{BBox}_t)^2 + d(\mathbf{p}^i_{\max}, \text{BBox}_t)^2, \tag{2}$$

where $d(\mathbf{p}, \mathrm{BBox}) = 0$ if $\mathbf{p} \in \mathrm{BBox}$, otherwise, $d$ is the Euclidean distance from $\mathbf{p}$ to the box boundary. The total loss is defined as

$$L(q_{n+1}(q_0)) = \sum_{i=1}^{N} l_i, \tag{3}$$

where $N$ is the total number of objects in the scene.

Direct gradients of $q_{n+1}$ with respect to $q_0$ cannot be obtained by differentiating the static equilibrium constraint $f(q_{n+1}) = 0$, since $q_0$ serves only as the initial guess and is not part of the constraint. Differentiating through the nonlinear solver is also prohibitively expensive. Instead, we adopt an artificial time-stepping formulation Fang et al. (2021), in which the quasi-static system gradually evolves toward equilibrium across intermediate states. This enables efficient backpropagation from $q_{n+1}$ to $q_0$ via implicit differentiation at each step. See Appendix B for more details on our forward simulation method and the derivation of differentiation.

## 4 EXPERIMENTAL RESULTS

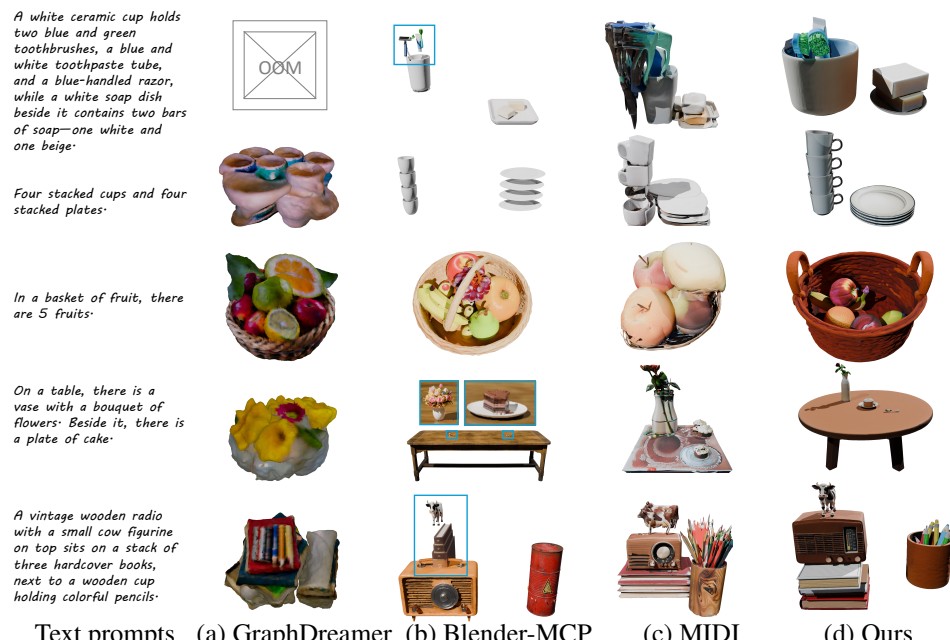

Figure 3: **Comparison to baseline methods.** The scenes are generated from our text prompts. OOM indicates out of memory.

### 4.1 COMPARISON

#### 4.1.1 BASELINES

We compare our method against three baselines: GraphDreamer Gao et al. (2024), MIDI Huang et al. (2024), and Blender-MCP or ahujasid (2025). Both GraphDreamer and Blender-MCP take text prompts as input, while MIDI uses a reference image as input. To ensure a fair comparison, we provide MIDI with our scene reference image as their input.

#### 4.1.2 DATASET

Since there is no standard benchmark for general scene generation, we construct our own test dataset consisting of 18 text prompts. Among them, 3 prompts are taken from MIDI, and 2 prompts are from GraphDreamer. Additionally, we use an LLM to generate 13 new text prompts spanning diverse

|  | Clip Score ↑ | VQA Score ↑ | Displacement ↓ | Pene. Ratio ↓ | Phys. Score ↑ |
|---|---|---|---|---|---|
| GraphDreamer | 27.53 | 0.46 | 0.25 | 61.72 | 40.0 |
| Blender-MCP | 28.93 | 0.56 | 1.03 | 14.78 | 47.7 |
| MIDI | 29.68 | 0.63 | 0.69 | 110.80 | 62.7 |
| Raw layout | 29.88 | 0.64 | 0.81 | 14.11 | 65.5 |
| Scene Init. | 30.77 | **0.70** | 2.91 | **0** | 34.2 |
| Ours | **31.79** | 0.68 | **0** | **0** | **88.5** |

Table 1: **Quantitative Evaluation.** Our method achieves the highest semantic consistency with input text prompts among all baselines, and is the only method that achieves perfect physical stability and non-intersection. We also ablates results without layout initialization and optimization, shown as raw layout.

scenes. These prompts describe physical interactions between objects, including a stack of books and a basket of fruits. Additional 3D results generated by our method, in comparison with the baselines, along with corresponding text prompts and reference images, can be found on visualization website[1]. Beyond these comparisons, we also present 12 more examples produced by our method in Appendix E.

### 4.1.3 EVALUATION METRICS

We evaluate our generated scenes using five metrics: CLIP Score Radford et al. (2021), VQA Score Lin et al. (2024), Simulated Scene Displacement (D), the Ratio of Penetrating Triangle Pairs (R), and a Physical Plausibility Score. Together, these metrics measure semantic consistency, physical stability, interpenetration, and overall physical plausibility. Details of all metrics are provided in Appendix F.

### 4.1.4 PERFORMANCE AND DISCUSSIONS

In Figure 3, we compare PAT3D with baseline methods on five general scenes involving complex object interactions. Additional comparisons with four scenes highlighted in the MIDI and Graph-Dreamer are provided in Appendix E. Importantly, in PAT3D, reference image serves only to extract the complex spatial relations implied in the text prompt; the final scene does not need to remain visually consistent with the reference image.

GraphDreamer struggles to scale to larger scenes because it jointly optimizes both object geometry and scene layout through Score Distillation Sampling (SDS) Poole et al. (2023), which is highly resource-intensive. Moreover, GraphDreamer exhibits weak understanding of spatial relations in text prompts. As shown in the second, fourth, and fifth scenes of Figure 3, it often ignores spatial constraints, leading to chaotic object arrangements. Blender-MCP generates layouts with little physical realism. In the first and second scenes of Figure 3, the razor and toothbrush float above the cup, and the plate is suspended in mid-air. It also produces objects with unrealistic scales: in the fourth scene, the cake and vase appear disproportionately small compared to the table. MIDI encounters difficulties when handling scenes with complex object contact, as seen in the first, second, and fifth scenes of Figure 3, objects often appear in irregular yet tightly packed configurations. Although interpenetration is avoided, the resulting layouts are cluttered potentially because MIDI generates the entire scene in a single step, the quality of individual objects is compromised.

By contrast, our method decomposes the scene generation process, iteratively creating objects to ensure high-quality results. Leveraging VLM-based guidance together with physics simulations, PAT3D produces 3D scenes that are both physically realistic and semantically consistent, even in scenarios with complex object interactions. Quantitative comparisons are shown in Table 1. Compared to baseline methods, which frequently suffer from object intersection and floating artifacts that undermine physical plausibility, our approach consistently produces stable, penetration-free arrangements. Our method also achieves the highest semantic consistency with the input text prompts.

---

[1]https://3dsim-baseline-visualization.netlify.app/

### 4.2 APPLICATION

Our generated simulation-ready scenes can be directly imported into a simulator for downstream applications. We demonstrate two such applications: scene editing and robotic manipulation.

#### 4.2.1 SCENE EDITING

We demonstrate a scene editing application enabled by our framework, which supports interactive manipulation while preserving the physical plausibility of the entire scene, including object addition and deletion. By leveraging our physics-based simulation backend, the edited scene converges to a force-equilibrium state without mesh intersections. Figure 4 highlights an example showcasing object addition and deletion, and animation results are provided in the supplementary material.

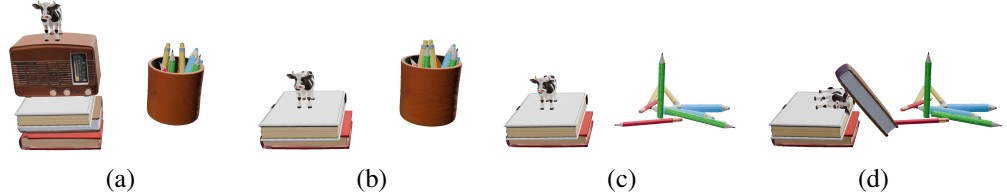

(a)     (b)     (c)     (d)

Figure 4: **Scene editing.** We demonstrate the equilibrium state after addition and deletion operations: (a) initial scene, (b) deleting a book at the bottom, (c) deleting the pen holder, (d) adding a book on top.

#### 4.2.2 ROBOTIC MANIPULATION

Our generated scenes can be directly imported into a simulator to validate robotic manipulation policies. In Figure 5, we present two illustrative examples, a failed grasp and a successful grasp, with video sequences provided in the supplementary material. Robotic manipulation applications impose unique requirements on scene generation: objects must be consistently positioned and free of interpenetrations. Our framework satisfies these requirements, ensuring that the generated scenes are well-suited for reliable policy evaluation.

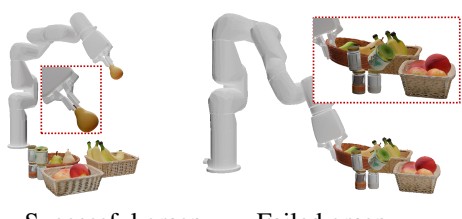

Successful grasp   Failed grasp

Figure 5: **Policy evaluation for robotic manipulation.** Example of a successful and a failed grasp where the attempted action causes objects to topple.

### 4.3 ABLATION STUDY

We first qualitatively illustrate the impact of our layout initialization module and simulation-in-the-loop optimization module. As shown in Figure 6(a), while the spatial relations between objects extracted directly from the depth map are generally reasonable, the scene still suffers from significant interpenetration: books intersecting with one another and the pen protruding its holder. By contrast, after applying our proposed layout initialization based on the scene tree, we obtain an intersection-free layout shown in Figure 6(b), where projections of each object along gravity direction typically lies in the projections of their containers or supporters, thereby satisfying physical dependency constraints along the gravity direction.

Nevertheless, simply enforcing physical dependencies before simulation does not ensure that the resulting simulated scene would still satisfy the intended semantics. In Figure 7(a), a stack of irregular blocks collapses after simulation due to an unbalanced center of mass. In contrast, Figure 7(b) demonstrates that, by further optimizing the initial layout through our simulation-in-the-loop optimization, the simulated scene converges to a stable configuration of stacked blocks that better reflects the semantics.

We further quantitatively evaluate the effectiveness of our method by computing semantic and physical metrics on both the depth-aligned layout without our layout initialization and optimization (denoted as *raw layout*), the layout from our scene initialization module (denoted as *Scene Init.*), and

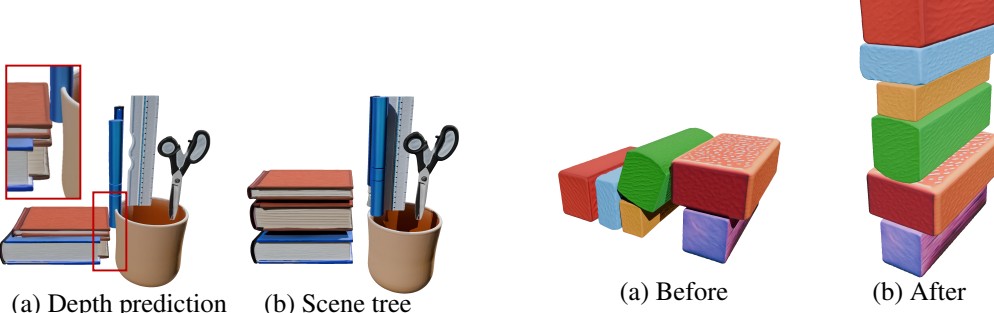

| (a) Depth prediction | (b) Scene tree | | (a) Before | (b) After |

Figure 6: **Layout initialization w/o and w/ scene tree.** Layouts obtained from depth prediction (a) without and (b) with adjustment based on the scene tree. (Text prompt: *"...a neatly stacked pile of three books..."*. See Appendix D for the complete prompt.)

Figure 7: **Layout optimization.** Simulated layouts from initial layout (a) without and (b) with further optimization using differentiable simulation. (Text prompt: *"a stack of colorful wooden blocks..."*. See Appendix D for the complete prompt.)

compare them with our final output in Table 1. Our layout initialization removes penetration by sacrificing physical stability, but it enables applying our layout optimization to consistently improve all metrics. The gains in semantic consistency metrics are relatively smaller, as they primarily depend on visual appearance factors such as geometry and texture.

## 5 CONCLUSION

We presented **PAT3D**, a physics-augmented framework for text-to-3D scene generation that integrates vision-language reasoning with differentiable rigid body simulation. By decomposing the generation process into interpretable stages – object and relation extraction, layout initialization, and physics-guided layout optimization – our method produces 3D scenes that are not only semantically meaningful but also physically plausible and simulation-ready. Through extensive experiments on diverse, contact-rich scenes, we demonstrated that PAT3D achieves superior physical realism compared to existing approaches. We believe PAT3D represents a step forward in bridging high-level scene understanding with low-level physical reasoning. We hope this work inspires further research in physically grounded, controllable, and editable 3D scene generation.

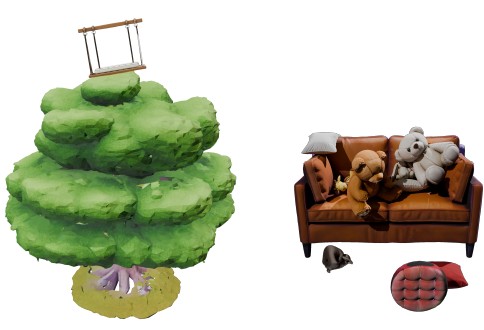

Figure 8: **Failure case 1.**
*"A swing hanging from a tree."*

Figure 9: **Failure case 2.**
*"A brown leather sofa decorated with plush toys, ..."*

**Limitations and Future Work** We present two representative failure cases in Figure 8 and Figure 9. While PAT3D handles most common physical dependencies, certain subtle relations remain challenging. In Figure 8, the concept of "hanging" in the prompt "A swing hanging from a tree" is misinterpreted, whereas a physically correct configuration would require the swing to be suspended from two specific attachment points. Extending our system to a broader set of spatial relations and larger-scale scenes are both promising directions for future work. One potential avenue is to incorporate path planning techniques and adopt a hierarchical optimization strategy to manage spatial complexity more effectively.

Our simulation-in-the-loop optimization is solved using a local optimizer. Although the objective reliably decreases, it does not guarantee convergence to the global optimum corresponding to perfect

semantic alignment. This limitation appears in the failure case of Figure 9, where all stuffed toys should ideally rest on the sofa according to the prompt. However, due to the highly crowded configuration, the optimizer provides a suboptimal outcome: it reduces the number of fallen toys from three to one, yet one gray toy still remains on the floor. As the first work to incorporate differentiable simulation into 3D scene generation, we view the exploration of global optimization strategies as an exciting direction for future improvements, further expanding the possibilities opened by PAT3D.

## 6    ACKNOWLEDGMENTS

Minchen Li was supported in part by the Junior Faculty Startup Fund of Carnegie Mellon University and a gift from Genesis AI. Jun-Yan Zhu was supported by the Packard Foundation and a Cisco Research Grant. Taku Komura was supported by the Innovation and Technology Commission of the HKSAR Government under the ITSP-Platform grant (Ref: ITS/335/23FP).

The authors would like to thank Yuezhi Yang, Yuanhao Wang, Lingting Zhu, Donglai Xiang, Kangle Deng, Liwen Wu, Yang Zheng, Yehonathan Lipman, Gaurav Parmar, Maxwell Jones, and Kevin You for their insightful discussions and feedback. We also extend our gratitude to Xinyu Lu for his professional support and key contributions to the development of Libuipc.

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

# A  PSEUDO-CODE OF BUILDING SCENE TREE

---

**Algorithm 1** Build Scene Tree

---

**Require:**
    Scene objects $\mathcal{O}$
    Root node $\mathscr{G}$ (ground)
**Ensure:**
    Hierarchical scene tree $\mathcal{T}$;
  1: Initialize tree $\mathcal{T}$ with root node $\mathscr{G}$;
  2: Mark all objects in $\mathcal{O}$ as unvisited;
  3: **Procedure** BuildSceneTree($n$):
  4: **for** $o \in \mathcal{O}$ where $o$ is unvisited **do**
  5:    **if** $o$ is in contact with $n$ **and** $o$ has an physical dependency relation with $n$ **then**
  6:       Add $o$ as a child of $n$ in $\mathcal{T}$;
  7:       Mark $o$ as visited;
  8:       **Call** BuildSceneTree($o$);
  9:    **end if**
10: **end for**
11: **Call** BuildSceneTree($\mathscr{G}$);

---

# B  DIFFERENTIABLE SIMULATION DETAILS

## B.1  FORWARD SIMULATION

We model each object in the scene as a stiff affine body Lan et al. (2022), where any point on the object with an initial position $\bar{\mathbf{x}}_{\text{init}}$ undergoes an affine transformation to its current position $\mathbf{x} = \mathbf{A}\bar{\mathbf{x}}_{\text{init}} + \mathbf{p}$, where $\mathbf{A} \in \mathbb{R}^{3\times3}$ is a transformation matrix and $\mathbf{p}$ is a translation vector. Together, they define the degrees of freedom (DOFs) of the object as $\mathbf{q} \equiv [\mathbf{p}, \mathbf{A}] \in \mathbb{R}^{3\times4}$.

To simulate the motion and contact of the objects, we employ a custom GPU-optimized affine body dynamics (ABD) simulator based on Huang et al. (2025a). The simulator solves for the configuration $q_{n+1} \in \mathbb{R}^{12N}$ at time step $n+1$, formed by flattening and stacking the DOFs of all $N$ objects, from the configuration $q_n$ at the previous time step via:

$$M(q_{n+1} - \tilde{q}_n) + \Delta t^2 \left( \nabla\Psi(q_{n+1}) + \nabla B(q_{n+1}) + \nabla D(q_{n+1}, q_n) \right) = 0. \tag{4}$$

Here, $M$ is the mass matrix, and $\tilde{q}_n = q_n + \Delta t^2 g$ is the predictive state used in artificial time stepping, which omits velocity. $\Delta t$ denotes the simulation time step, and $g$ is the gravitational acceleration. The potential $\Psi$ models stiff elasticity to preserve object shape, $B$ is a barrier potential enforcing non-penetration constraints, and $D$ is a semi-implicit friction potential following Li et al. (2020).

## B.2  BACKPROPAGATION

To optimize the initial layout $q_0$, we compute the gradient of the loss function $L$ w.r.t $q_0$ using the chain rule:

$$\frac{dL}{dq_0} = \left(\frac{\partial q_1}{\partial q_0}\right)^\top \left(\frac{\partial q_2}{\partial q_1}\right)^\top \cdots \left(\frac{\partial q_n}{\partial q_{n-1}}\right)^\top \left(\frac{\partial q_{n+1}}{\partial q_n}\right)^\top \frac{dL}{dq_{n+1}}. \tag{5}$$

Here, $\frac{dL}{dq_{n+1}}$ can be directly computed at the target step $n+1$ or automatically obtained via PyTorch Paszke (2019). The key step lies in computing $\frac{\partial q_{n+1}}{\partial q_n}$, which we derive using implicit differentiation. Rewriting Equation 4 yields:

$$q_{n+1} = q_n + \Delta t^2 M^{-1} \left[ f(q_{n+1}) + Mg \right], \tag{6}$$

where $f(q_{n+1}) = -\nabla\Psi(q_{n+1}) - \nabla B(q_{n+1}) - \nabla_{q_{n+1}} D(q_{n+1}, q_n)$. Differentiating both sides of Equation 6 with respect to $q_n$ and isolating the derivative yields:

$$\frac{\partial q_{n+1}}{\partial q_n} = \left[ I - \Delta t^2 M^{-1} \frac{\partial f(q_{n+1})}{\partial q_{n+1}} \right]^{-1} \left[ I - \Delta t^2 M^{-1} \frac{\partial^2 D(q_{n+1}, q_n)}{\partial q_{n+1} \partial q_n} \right]^{-1}. \tag{7}$$

Substituting Equation 7 into the chain rule expression in Equation 5 allows us to compute the full gradient $\frac{dL}{dq_0}$, which we use to update the initial layout. Both forward simulation and backpropagation are fully GPU-accelerated for computational efficiency.

## B.3 PHYSICAL AND ALGORITHMIC PARAMETERS

All simulations are performed using a differentiable rigid body simulator with the following physical and algorithmic parameters. They are generally applicable to rigid body scenes and only a few of them needs tuning. We define the mms, i.e. the mean mesh size, to be the average of the longest side of the bounding box of all meshes.

Fixed parameters:

- **Friction coefficient**: A Coulomb friction coefficient of $0.2$ is used for all contact interactions.
- **Normal contact stiffness**: The penalty-based normal-force model uses an effective stiffness of $1.0\,\mathrm{GPa}$, corresponding to rigid-body behavior.
- **Gravity**: Standard Earth gravity $\mathbf{g} = (0, -9.8, 0)\,\mathrm{m\,s^{-2}}$ is used.
- **Time step** ($\Delta t$): The simulation is integrated using a time step of $0.03\,\mathrm{s}$.
- **Newton solver tolerance**: In each time step, Newton's method is applied to solve the time integration problem, and the termination criteria is the velocity residual falls below $0.1\,\mathrm{m\,s^{-1}}$.
- **Maximum number of frames**: This specifies the duration of the simulation. We use 300 frames for all the scenes.
- **Optimization learning rate**: We use a learning rate of $0.001$ for simulation-in-the-loop optimization.
- **Maximum optimization epochs**: Differentiable simulation is allowed up to 50 optimization iterations.

Tunable paramters:

- **Contact distance threshold**: A threshold of $0.01$mms is used for collision handling in most cases. Two objects are considered in contact when their distance falls below this value. We use a value of $5 \times 10^{-4}$ for the stackedblocks scene to capture the intricate balancing behavior.
- **Friction velocity threshold**: Relative velocities below $0.01$mms are modeled to generate static friction forces for most cases. We use a value of $10^{-5}$ for the stacked blocks scene.
- **Optimization frame interval**: We compute semantic losses every 10 frames by defult and accumulate it during the simulation. This parameter are set according to the frequency of contact events in each simulation. Most of our examples achieve perfect semantic alignment at the first optimization iteration, and thus no tuning needed.

## B.4 TIMING AND MEMORY CONSUMPTION

All experiments are conducted on a single NVIDIA A5000 GPU. The average end-to-end time to generate a scene is 1632 seconds. The main computational cost arises from object generation, which requires an average of 762 seconds per scene. During the simulation-in-the-loop stage, each optimization iteration takes approximately 30 seconds, and we use 50 iterations in total, selecting the solution with the lowest objective value.

For all generated scenes, the simulation fits within 8 GB of GPU memory. In practice, memory usage scales with the size of the contact graph, which depends on the geometric complexity of the objects and the number of active contacts in the scene. Our formulation relies on sparse matrix representations for both system dynamics and contact operators, which helps maintain a moderate memory footprint.

Given ongoing improvements in GPU hardware and simulation techniques, we do not anticipate efficiency and memory to be the limiting factors for typical applications.

## C  IMPLEMENTATION DETAILS

We implement our proposed algorithm in Python on Ubuntu 20.04. In the layout optimization, we leverage Libupic Huang et al. (2025a) as a simulation platform. The optimization is performed using ADAM Kingma (2014). Scenes that are already semantically consistent after the first simulation are not optimized. Our algorithm is deployed and run on a single NVIDIA RTX 4090 GPU. All our baselines were tested on the NVIDIA A6000 GPU.

## D  TEXT PROMPT USED IN ABLATION STUDY

The complete text prompt used in our ablation study are as follows.

- Figure 6: *"On the left side, there is a metallic cylindrical pen holder containing two black pens, a wooden ruler, and a pair of gray-handled scissors. On the right side, there is a neatly stacked pile of three books with red covers and visible pages. The items are placed on a light wooden surface, and the background is plain white, creating a bright and simple composition."*

- Figure 7: *"a stack of colorful wooden blocks arranged vertically, featuring red, blue, yellow, green, orange, and purple pieces, balanced on a flat surface."*.

- Figure 9 *"A brown leather sofa decorated with plush toys, including two large teddy bears, a gray elephant, a white rabbit, a yellow giraffe, and two throw pillows, sits in a cozy room with two round burgundy floor cushions in front."*

## E  MORE EXAMPLES

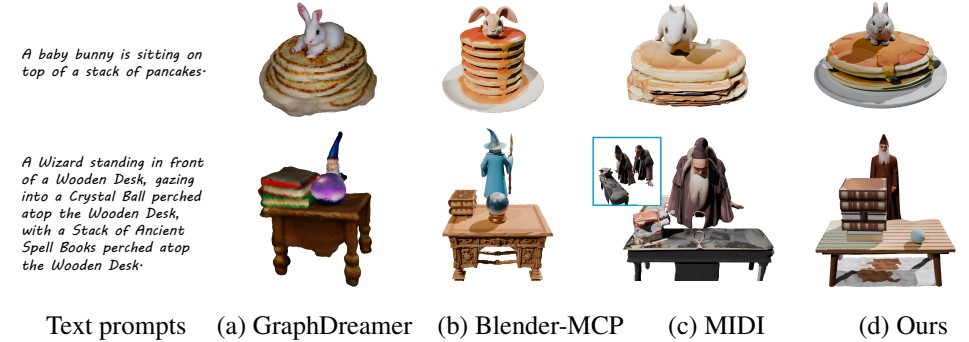

Figure 10: **Comparison of generated scenes from text prompts used in GraphDreamer** Gao et al. (**2024**).

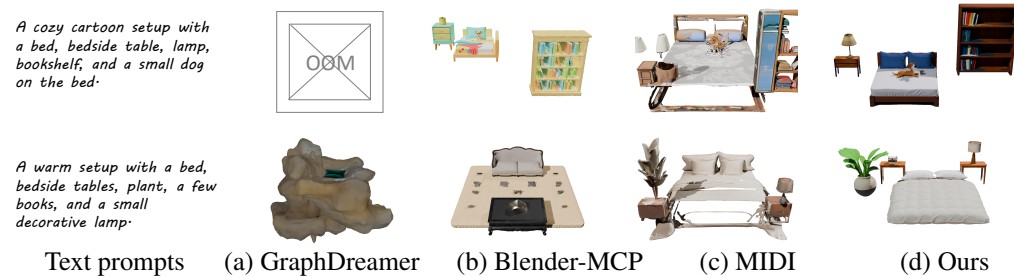

Figure 11: **Comparison of generated scenes from text prompts used in MIDI** Huang et al. (**2024**).

In addition to the 18 text prompts used for comparison with the baseline, we further tested our algorithm on 12 additional examples, as shown in Figure 12. All of these prompts yielded semantically accurate and physically stable results.

## F    METIRCS

**Clip score and VQAScore**    These two metrics measure the semantic similarity between the rendered scene images and the corresponding input text prompt. Specifically, we render the scene from 18 viewpoints by sampling three depression angles (0°, 20°, and 45°) and six evenly distributed horizontal angles. These rendered images are then used to compute the Clip score and VQAScore.

**Simulated Scene Displacement (D)**    This metric computes the normalized average displacement of object vertices in the scene before and after applying a simulation. To ensure consistency across different baselines, we first normalize all scenes such that their bounding box diagonal length equals 2. We then compute the displacement of every vertex over all simulated frames and aggregate these values into a single scalar quantity. Finally, this scalar is normalized by the total number of vertices and by the scene's diagonal length. Formally, let $v_j^{(t)}$ denote the position of vertex $j$ at simulation frame $t$, and let $V$ and $T$ denote the number of vertices and frames, respectively. The metric is defined as the average per-vertex displacement normalized by the scene diagonal length $l$:

$$D = \frac{1}{Vl} \sum_{j=1}^{V} \sum_{t=1}^{T} \left\| v_j^{(t)} - v_j^{(t-1)} \right\|, \tag{8}$$

**Ratio of Penetrating Triangle Pairs (R)**    This metric quantifies the extent of penetration between objects in the scene, serving as an indicator of the scene's geometric correctness. We first normalize all scenes such that their bounding box diagonal length equals 2. We then remesh each object using fTetWild Hu et al. (2020), setting the target triangle edge length to 0.05 times the scene diagonal. After normalization, we compute $R$ as the ratio between an approximated total length of intersection contours and the length of the scene diagonal $l$: $R = \frac{(T_p - \sum_{i=1}^{N} T_{p,i}) l_e}{l}$, where $T_p$ is the total number of penetrating triangle pairs in the scene, $T_{p,i}$ is the number of self-penetrating triangle pairs in object $i$, which is excluded, and $l_e$ is the average edge length of all object meshes.

**Physical Plausibility Score**    This VLM-based metric evaluates the physical plausibility of a generated scene by asking a GPT model to score the realism of object contacts and physical relationships in the rendered image. Specifically, we use the following prompt:

> "The semantic meaning of this scene is: '...'. Please evaluate whether the physical relationships in this image are reasonable and whether the contacts between objects are physically realistic. Give this scene a physical plausibility score from 0 to 100."

## G    MORE COMPARISON

We qualitatively compare our method with the baselines on the text prompts previously presented in MIDI and GraphDreamer, as shown in Figure 11 and Figure 10, respectively. For the comparison with MIDI, since it requires an image as input, we first generate images from the provided text prompts in their paper and use these as MIDI's inputs. For the other baselines, we directly use the text prompts. For our method, we use the same text prompts while ensuring that the reference image is consistent with the input image used for MIDI.

## H    COMPARISON WITH LAYOUT-YOUR-3D

Using the same text prompt as in Layout-Your-3D Zhou et al. (2025a), we compare a generated scene to highlight our improved physical plausibility. In Figure 13(a), PAT3D produces physically accurate contact. In contrast, in Figure 13(b), taken from the Layout-Your-3D paper, the basketball intersects the sofa, resulting in geometry penetration that precludes intersection-free simulation.

## I  DISCLOSURE

We made use of LLMs to polish writing. We made sure that our input text to LLMs will not be used for training purposes.

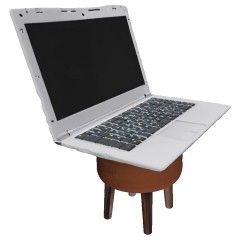

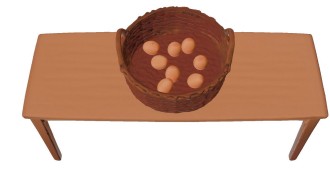

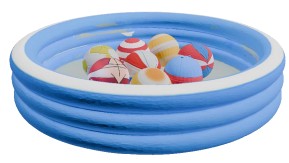

"A silver laptop with its screen turned off is placed on a small round wooden stool with three legs."

"A table with a basket of eggs on it."

"A small inflatable blue-and-white kiddie pool is filled with eight colorful beach balls with red, yellow, and white panels float on the surface. ."

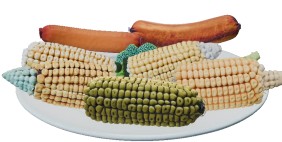

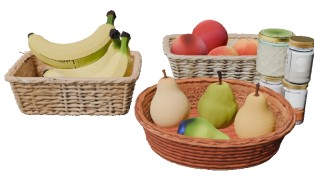

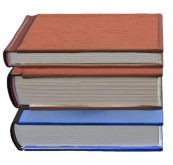

"A round white plate holds a simple arrangement of food: two brown sausages placed side by side, several pieces of bright yellow corn on the cob cut into chunks, and three green broccoli florets."

"Four woven baskets arranged in a row, each containing different fruit: pears, bananas, oranges, and apples, alongside two stacks of jam jars."

"There is a white cylindrical holder containing several stationery items, including a pair of black-handled scissors, a white ruler, a blue pen, and a mechanical pencil. To its left lies a neatly stacked pile of three hardcover books in earthy tones of orange, beige, and blue."

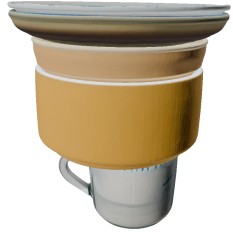

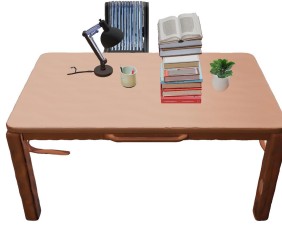

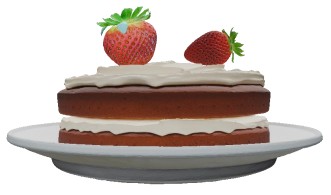

"A transparent glass mug supports a stack of two beige bowls and two beige plates, with a silver fork standing upright on the topmost plate."

"On the desk, a black desk lamp sits on the left, accompanied by a white mug filled with pencils. The right side is dominated by a tall stack of hardcover books, with an open book on top, while a small potted plant adds a touch of greenery on the far right. A black leather office chair is centered behind the desk."

"A small chocolate cake topped with a swirl of white whipped cream and decorated with two fresh red strawberries, served on a round white plate placed on a wooden table."

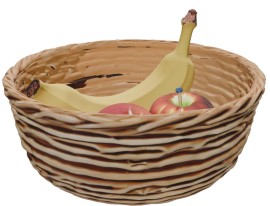

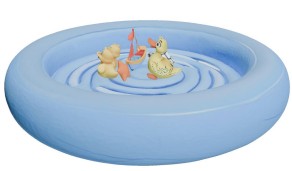

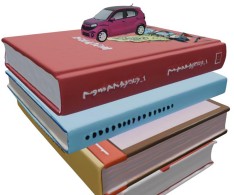

"A fruit basket containing one banana and two apple."

"An inflatable swimming pool with toy ducks, a boat, and a starfish toy."

"A toy car on four books."

Figure 12: More results of our method.

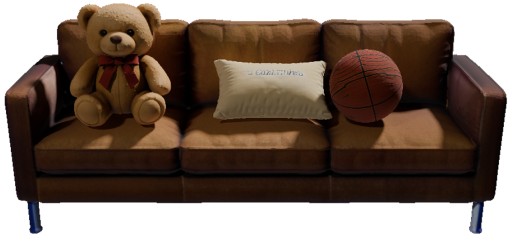 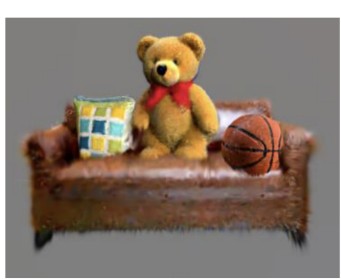

PAT3D                                    Layout-your-3D

Figure 13: Comparison between PAT3D and Layout-your-3D. (Text prompt: *"A brown sofa with a cushion, a teddy bear, and a basketball on it"*)

