# OpenReview forum: "PAT3D: Physics-Augmented Text-to-3D Scene Generation"
_ICLR.cc/2026/Conference — ICLR 2026 Poster_

### Official Review · Reviewer_kPCZ · 2025-10-30

**Soundness:** 2
**Presentation:** 3
**Contribution:** 3
**Rating:** 6
**Confidence:** 4

**Summary:**

This paper introduces PAT3D, a physics-augmented framework for text-to-3D scene generation that addresses the critical gap of physical plausibility in multi-object scene synthesis. The technical approach integrates vision-language models with differentiable rigid body simulation to produce scenes that are both semantically consistent and physically stable. The method decomposes the generation pipeline into three stages: object and spatial relation extraction via VLMs, physics-aware layout initialization using hierarchical scene trees, and simulation-in-the-loop optimization. Experiments demonstrate improvements over existing methods in physical stability metrics and semantic consistency.

**Strengths:**

- This paper focuses on a meaningful and underexplored problem. The lack of physical plausibility in text-to-3D scene generation is important for downstream applications in robotics and simulation.
- Demonstrates practical applications in scene editing and robotic manipulation tasks.
- Clear experimental validation showing zero interpenetration and stable equilibrium states.

**Weaknesses:**

- Limited video demonstrations make it difficult to fully assess the dynamic behavior and stability of generated scenes under various conditions. The supplementary materials would benefit from more extensive video results. Currently there are only 2 examples.
- Although the paper claims the first work on physics-plausible scene generation, I'm not sure how different the problem setting is from CAST. From the demo examples of CAST, they also have physics simulation to make sure objects are placed in a stable and non-penetrating way. How is the proposed work different from it?

**Questions:**

Generally I think this is an important problem and the results look interesting. But even in the two examples (which are very limited), the simulation looks a bit unrealistic. So I'm not sure if the technical approach is really addressing the challenges stated in the paper.

Also, I'm not fully convinced why the work is different from CAST.

---

> ### Author Response · Authors · 2025-11-22
>
> We appreciate your careful evaluation and are encouraged by the positive recognition of the problem’s significance, the demonstrated applications, and the clarity of the experimental results on physical stability and semantic consistency. Below, we address the concerns and questions raised.
>
> &nbsp;
>
> **Weakness 1**: Limited video demonstrations make it difficult to fully assess the dynamic behavior and stability of generated scenes under various conditions. The supplementary materials would benefit from more extensive video results. Currently there are only 2 examples.
>
> **Weakness 2**: But even in the two examples (which are very limited), the simulation looks a bit unrealistic. So I'm not sure if the technical approach is really addressing the challenges stated in the paper.
>
> &nbsp;
>
> Thank you for the great suggestion. In response, we have added two additional videos in the supplementary material, showing physics-based animations created using our generated scenes. In the animations, the scene remains static before we throw an additional object into it to destroy the established structures, demonstrating the stability and realistic dynamic behaviors achievable using our generated scenes under various conditions.
>
> Regarding the reviewer’s concern about simulation realism, the unusual dynamics observed in the visualization of our simulation-in-the-loop optimization are due to the use of **artificial time stepping** [1], which zeros out the velocity after each time step solve to ensure that the static equilibrium can be reached more efficiently. This choice is intentional; it avoids unnecessary bouncing effects while also making the motion look unnatural, but it still guarantees penetration-free and accurate final static equilibrium state.
> In the newly added animation videos, we used the **standard dynamic simulation** to capture the realistic dynamics. We would be happy to provide more animations if requested.
>
> &nbsp;
>
> *[1] Fang Y, Li M, Jiang C, Kaufman DM. Guaranteed globally injective 3D deformation processing. ACM Transactions on Graphics. 2021 Aug;40(4).*
>
> &nbsp;

---

> ### Author Response · Authors · 2025-11-22
>
> **Question 1**: how different the problem setting is from CAST. From the demo examples of CAST, they also have physics simulation to make sure objects are placed in a stable and non-penetrating way. How is the proposed work different from it?
>
> &nbsp;
>
> Thank you for the insightful comment. We would like to clarify that the key difference between our method and CAST is that **our method achieves physical realism and a strict intersection-free guarantee, while CAST cannot**.
>
> **Our work is the first to incorporate differentiable simulation inside the optimization loop for 3D scene generation**, whereas CAST focuses more on simultaneous generation of multiple objects rather than making the results simulation-ready and physically realistic, and thus they only uses a simulation-motivated “physics-aware correction” module without performing an actual physics-based simulation. In their module, CAST simply applies SDF-based penalties: it samples surface points, penalizes negative SDF values (penetration), and encourages at least one zero-SDF point for contact. As the CAST authors explicitly acknowledge:
>
> *“We argue that our optimized results can serve as a reliable initialization for subsequent physical simulations.”*
>
> *“Note that our approach does not model full dynamics. For example, an object may not remain stable in its current pose over time.”*
>
> Consequently, CAST cannot guarantee **strictly intersection-free** results in cluttered scenes. For instance, in CAST Figure 1 (second row, third and fourth columns), noticeable penetrations remain. In contrast, our pipeline enforces exact non-penetration via differentiable simulation, which is essential for producing simulation-ready scenes that can be loaded into a high-accuracy, intersection-free simulator – an important motivation for embodied AI applications.
>
> Moreover, CAST does not guarantee **physically realistic contact configurations**. Our simulation-in-the-loop approach yields physically realistic contact because the solver explicitly resolves force balance. This distinction is crucial for downstream tasks where the accuracy of physical interactions matters. One concrete example is the stacked-blocks scene in our Figure 1. The blocks have irregular shapes, and achieving a stable final arrangement requires subtle **adjustments to the center of mass**, which is only achievable through simulation-based optimization. CAST’s simplified physical-principle losses cannot capture such delicate balance conditions and therefore cannot achieve comparable physical plausibility.
>
> To the best of our knowledge, our method is the first to explicitly address the key challenges that make applying simulation difficult in 3D scene generation – partial scenes, imperfect geometries, and large initial penetrations – and to demonstrate that differentiable simulation provides a practical path toward physically grounded scene generation.
>
> &nbsp;
>
> Once again, thank you for your insightful review. We have incorporated the suggested changes into our paper. Please let us know if you have any additional questions or concerns. We are happy to provide further clarifications or modifications as needed.

---

### Official Review · Reviewer_etTa · 2025-10-31

**Soundness:** 3
**Presentation:** 2
**Contribution:** 2
**Rating:** 4
**Confidence:** 3

**Summary:**

The paper introduces PAT3D, a framework for generating 3D scenes from textual prompts, with emphasis on physical plausibility and simulation-readiness.
This work uses a 3D Objects and Spatial Relation Extraction module by generating reference images, a Layout Initialization module with the help of point clouds, scenetree, and a Layout Optimization module with self-defined physical constraints to generate simulatable and physically plausible 3D scenes.
In short: PAT3D bridges text→3D scene generation with physics simulation, aiming not just for plausible appearance but also for physically stable, actionable 3D scenes.

**Strengths:**

- The visualization results are obviously better than previous works.
- The pipeline is intuitive and direct. With the help of depth and point cloud, this work is able to obtain precise 3D scene layouts and spatial relationships.
- Applying simulations in the process of 3D scene generation is an interesting method, which alleviates a lot of trouble in the nuisance 3D layout optimization process.

**Weaknesses:**

- The novelty of this work is relatively weak. The '3D OBJECT AND SPATIAL RELATION EXTRACTION' part also appears in [1], and the novel parts are the simulation instead of 3D scene optimization, which is a little bit weak.
- The use of rigid-body simulation and “simulation in the loop” likely requires non-trivial compute and careful tuning. The reproducibility, speed, and resource requirements might limit broader adoption.
- Too few quantitative results, since the paper features the simulation ability of this work, I guess some more metrics that can reflect the physical plausibility and the advantage for manipulations are needed.

[1]: Layout-your-3D: Controllable and Precise 3D Generation with 2D Blueprint

**Questions:**

- What is the success rate of this work? Will the simulation finally converge to a plausible point with multiple iterations? Or it is largely dependent on the initialization?
- Are there more failure cases? This failure case is not that typical, and I guess it can not reflect the major limitation of this work.
- How much time does it take to generate a single 3D scene with the simulation-in-the-loop? What is your edge over [1]?

[1]: Embodiedgen: Towards a generative 3d world engine for embodied intelligence

---

> ### Author Response · Authors · 2025-11-22
>
> We appreciate your constructive feedback. We are encouraged by the positive remarks highlighting the clarity of the pipeline, the improved visual quality, and the value of incorporating simulation to streamline 3D layout refinement. Below, we address the concerns and questions raised.
>
> &nbsp;
>
> **Weakness 1**: The novelty of this work is relatively weak. The '3D OBJECT AND SPATIAL RELATION EXTRACTION' part also appears in [1], and the novel parts are the simulation instead of 3D scene optimization, which is a little bit weak.
>
> &nbsp;
>
> We would like to clarify that the use of simulation in 3D scene generation is not as trivial as post-processing, and our contribution lies in two key technical components:
>
> 1. **A carefully designed, intersection-free, physics-aware scene initialization that is explicitly constructed to be simulation-ready (but not yet semantically accurate or physically stable, which would be addressed in the next component).**
>
> This step is non-trivial, as applying physical simulation to generated scenes typically faces significant challenges such as partial or missing geometry, imperfect object shapes, and large initial penetrations. Our initialization explicitly addresses these issues, making downstream simulation both feasible and reliable.
>
> 2. **A 3D scene optimization framework that integrates the differentiable simulator with the scene tree, which encodes the physical dependency structure of the scene.**
>
> Leveraging both differentiable simulators and a scene tree, we formulate an objective function within a barrier-based optimization framework that jointly enforces physical stability and semantic consistency.
>
> In contrast, prior scene generation works, such as Layout-your-3D, and related approaches cannot reliably yield high-quality results when simulation is directly applied. This is because:
>
> - **They cannot strictly guarantee non-intersection.**
>
> Or
>
> - **They cannot maintain semantic consistency after simulation** (e.g., irregular block stacks collapsing after dynamics), since their pose optimization does not involve simulation.
>
> We do not claim novelty in “3D object and spatial relation extraction”. However, our purpose and usage of this “input preparation module” differ fundamentally from prior work. Instead of optimizing scene layout directly from extracted relations, as done by previous methods, we extract the objects and the scene tree **specifically to support physics-based simulation**, allowing the simulator to naturally resolve contacts in a physically realistic way.
>
> &nbsp;

---

> > ### Author Response · Authors · 2025-11-22
> >
> > **Weakness 2**:  The use of rigid-body simulation and “simulation in the loop” likely requires non-trivial compute and careful tuning. The reproducibility, speed, and resource requirements might limit broader adoption.
> >
> > **Question 3**: How much time does it take to generate a single 3D scene with the simulation-in-the-loop?
> >
> > &nbsp;
> >
> >
> > To the best of our knowledge, our work represents the first integration of differentiable simulation into a 3D scene generation pipeline. We demonstrate that this integration yields substantial improvements in both physical plausibility, visual quality, and quantitative performance, outperforming prior methods [1, 2]. At the same time, we acknowledge that there is room for improvement, and we address R2’s valuable concerns point by point:
> >
> > - **Careful tuning**:
> >
> > The physical and algorithmic parameters used in our simulation-in-the-loop optimization are reported in the Appendix in the revised manuscript. Although many parameters exist, most of them are standard for rigid body contact simulation and thus are kept the same for all examples. Only the contact distance threshold, friction velocity threshold, and the optimization frame interval require tuning.
> > Here, the former two control the efficiency-accuracy trade-off, specifically, we used smaller values for the stacked blocks example to capture the intricate balancing behavior, while a larger value was used for all other cases to gain more efficiency. The optimization frame interval specifies the frequency we evaluate the loss function during the simulation. It is set according to the frequency of contact events during the simulation. Still, most of our examples achieve perfect semantic alignment at the first optimization iteration, and thus no tuning needed. In the future, it would certainly be meaningful and interesting to explore more convenient and intuitive parameter control for simulation-in-the-loop optimization.
> >
> > - **Reproducibility**:
> >
> > Our method is fully reproducible. It is built on top of an open-source simulation library [3], and we will release all data and scripts necessary to reproduce our examples.
> >
> > - **Speed and resource requirements**:
> >
> > We evaluate the runtime on a single NVIDIA A5000 GPU. The average time to generate one scene is 1632 seconds, where the major bottleneck is object generation, which takes 762 seconds on average per scene. For the simulation-in-the-loop stage, each simulation-based optimization iteration costs approximately 30 seconds, and the number of optimization iterations is set to 50, where the result in the iteration with the smallest loss is selected as our final solution.
> > For all our generated scenes, the simulation fits within 8 GB of GPU memory. In general, the memory footprint scales with the size of the contact graph, which is strongly correlated with the number of vertices/edges in the geometry and the number of active contacts (collisions) in the scene. Since our formulation is based on sparse matrix representations of the system and contact operators, together with the rapid growth of available GPU memory on modern hardware, we do not expect memory usage to be a practical bottleneck for typical applications.
> >
> > We also note that this is a rapidly advancing area – simulation speed and memory efficiency improve significantly each year, which will naturally benefit our framework moving forward. [4]

---

> > > ### Author Response · Authors · 2025-11-22
> > >
> > > **Weakness 3**: Too few quantitative results, since the paper features the simulation ability of this work, I guess some more metrics that can reflect the physical plausibility and the advantage for manipulations are needed.
> > >
> > > &nbsp;
> > >
> > > Thanks for the thoughtful comment. We evaluated physical plausibility using two metrics: **Simulated Scene Displacement (D)** and the **Ratio of Penetrating Triangle Pairs (R)**, which together capture the two essential aspects of physical correctness: **stability** and **intersection-free geometry**. For better clarity, we also update these two metrics with a normalized version in the revised manuscript. These properties also directly reflect a scene’s **suitability for manipulation**: intersections can induce unintended coupled motion during grasping, and unstable scenes often collapse under simulation, leading to task failure. In other words, an ideal manipulation-ready scene must be intersection-free and remain stable in a simulator.
> > >
> > > In addition, we introduce a **GPT-based physical plausibility score**, using the following prompt:
> > >
> > > *“The semantic meaning of this scene is: ‘xxx’. Please evaluate whether the physical relationships in this image are reasonable and whether the contacts between objects are physically realistic. Give this scene a physical plausibility score from 0 to 100.”*
> > >
> > > We report this **Physical Plausibility Score** in Table 1 in the revised manuscript. Across all baselines, **PAT3D obtains the highest score**, demonstrating its superior physical realism.
> > >
> > > We agree that there can be more requirements for 3D data to be simulation-ready, e.g. whether the mesh is watertight and formed by elements with good aspect ratio, etc. Since these are relatively orthogonal to scene generation and usually can be fixed by standard post-processing tools, we did not include them as additional metrics.
> > >
> > > &nbsp;
> > >
> > > **Question 1**: What is the success rate of this work? Will the simulation finally converge to a plausible point with multiple iterations? Or it is largely dependent on the initialization?
> > >
> > > **Question 2**: Are there more failure cases? This failure case is not that typical, and I guess it can not reflect the major limitation of this work.
> > >
> > > &nbsp;
> > >
> > > Thank you for the insightful comment. We clarify the robustness of our method from two perspectives – **physical accuracy** and **semantic alignment**.
> > >
> > > **1. Physical accuracy.**
> > >
> > > PAT3D is designed such that physically plausible outcomes are ensured **by construction**. Our optimization is carried out within a simulation-in-the-loop framework that uses barrier energy to enforce non-penetration and force balance throughout the process. Consequently, while semantic alignment is refined, physical correctness is preserved.
> > >
> > > **2. Semantic alignment.**
> > >
> > > Our **simulation-in-the-loop optimization always improves semantic alignment**, regardless of initialization. However, since our method currently uses local optimization, it cannot guarantee reaching the global semantic optimum in all cases. Exploring global optimization strategies such as particle swarm optimization and simulated annealing is an exciting avenue for future work.
> > > If we define success as having perfect semantic alignment, then, across the 30 test cases, our method achieves a high success rate of **28/30**. The two failure cases are shown in Fig. 8 and Fig. 9.
> > >
> > > - **Fig. 8**: This failure is due to a subtle physical dependency “a swing hanging from a tree”, which PAT3D cannot yet fully interpret. Correctly modeling such suspended, multi-point attachment relations remains a meaningful future work.
> > >
> > > - **Fig. 9**: Initially, three toys were mistakenly placed on the floor instead of the sofa. After optimization, only one remained off the sofa, with the others correctly repositioned. The final arrangement is semantically improved compared to the initial solution, though not perfectly matched to the target. This occurs because our simulation-in-the-loop optimization may converge to a **local** rather than a global optimum.
> > >
> > > We would like to clarify that we do not encounter failures such as program crashes or numerical instabilities. While perfect semantic alignment cannot be guaranteed, our approach consistently achieves strong semantic coherence while ensuring physical plausibility, which outperforms existing methods.
> > >
> > > &nbsp;

---

> > > > ### Author Response · Authors · 2025-11-22
> > > >
> > > > **Question 4**: What is your edge over [2]?
> > > >
> > > > Although **EmbodiedGen** also performs text-to-3D scene generation that can be imported into a simulator, its pipeline does not incorporate physics simulation for optimization. Their “Physics Restoration” module only performs **scale estimation** to ensure that object sizes within the scene are roughly consistent. However, it does not ensure intersection-free placement between objects and can still produce physically implausible contacts. Furthermore, it cannot ensure that objects remain stable or preserve the intended semantics once placed in a simulator under gravity, especially in scenes with complex contact relations. In contrast, our method explicitly addresses all of these issues through simulation-in-the-loop optimization.
> > > >
> > > > &nbsp;
> > > >
> > > > Once again, thank you for your insightful review. We have incorporated the suggested changes into our paper. Please let us know if you have any additional questions or concerns. We are happy to provide further clarifications or modifications as needed.
> > > >
> > > > &nbsp;
> > > >
> > > > *[1] Layout-your-3D: Controllable and Precise 3D Generation with 2D Blueprint*
> > > >
> > > > *[2] Embodiedgen: Towards a generative 3d world engine for embodied intelligence*
> > > >
> > > > *[3] libuipc: A Modern C++20 Library of Unified Incremental Potential Contact.*
> > > >
> > > > *[4] Lan L, Lu Z, Yuan C, Xu W, Su H, Wang H, Jiang C, Yang Y. JGS2: Near Second-order Converging Jacobi/Gauss-Seidel for GPU Elastodynamics. arXiv preprint arXiv:2506.06494. 2025 Jun 6.*

---

> > > > > ### Comment · Reviewer_etTa · 2025-11-26
> > > > >
> > > > > Thank you for your response! I think some of my concerns have been addressed. It is good to see another new metric is introduced. I think if it is possible, you can incorporate the discussions of the works like (EmbodiedGen, Layout-your-3d) into the paper (considering the page limit, it's okay to go with the appendix), and show the advantages of your work. Some visualization comparisons are better.
> > > > > Overall, I think your explanation is reasonable. I decided to raise my score to 6.

---

> > > > > > ### Author Response · Authors · 2025-12-03
> > > > > >
> > > > > > Thank you for the helpful suggestions and for recognizing our contributions. In the revision, we have added discussions of **EmbodiedGen** and **Layout-Your-3D** in the related work section and clearly explained our advantages over these methods. Due to time constraints, we have added one comparison in the appendix, and we will add more in the future revision.

---

### Official Review · Reviewer_D8i8 · 2025-11-01

**Soundness:** 3
**Presentation:** 3
**Contribution:** 3
**Rating:** 6
**Confidence:** 1

**Summary:**

PAT3D is a physics-augmented text-to-3D scene generation framework that overcomes common issues in geometry-only layouts (floating, unstable stacks, wrong supports) by integrating differentiable rigid-body contact simulation into the pipeline. From a text prompt, it synthesizes a reference image, generates segmented object meshes, and uses a VLM to infer inter-object physical dependencies, forming a hierarchical scene tree. It then constructs an intersection-free, physics-aware initialization by inserting small gravity-aligned gaps for support pairs, and lets objects settle via simulation before optimizing the layout with differentiable, artificially time-stepped dynamics for semantic and physical consistency. Experiments on contact-rich scenes show state-of-the-art visual quality, semantic alignment, and physical plausibility, with scenes remaining editable and directly usable for simulation-based robotics evaluation.

**Strengths:**

- More realistic physics: Differentiable rigid-body contact simulation reduces floating, interpenetration, and unstable stacks.
- Better semantic–layout alignment: A VLM infers object dependencies to build a scene tree that preserves text-described spatial relations.
- Stable, collision-free initialization: Physics-aware initialization with small gravity-aligned gaps improves convergence and avoids intersections.
- Optimizable and editable scenes: Layout is refined via differentiable simulation; outputs remain easy to edit and interact with.

**Weaknesses:**

- Scalability to very large scenes: Many-object, dense-contact scenes increase solver complexity (contact graph size), potentially impacting stability and runtime.
- Lack of quantitative study in ablation:   Measurable gains introduced by each component are necessary, making audience further understand the contributions.

**Questions:**

Please check Weaknesses

---

> ### Author Response · Authors · 2025-11-22
>
> We appreciate your time and thoughtful assessment of our work. We are encouraged by the positive recognition of the framework’s physical realism, semantic layout alignment, stable initialization strategy, and the practical value of producing scenes that remain optimizable and editable. Below, we address the concerns and questions raised.
>
> &nbsp;
> &nbsp;
>
>
> **Weakness 1**: Scalability to very large scenes: Many-object, dense-contact scenes increase solver complexity (contact graph size), potentially impacting stability and runtime
>
> &nbsp;
>
> Thank you for the insightful comment.
>
> **Scalability to large scenes.**
>
> Our work focuses on a fundamental challenge in 3D scene generation: ensuring physically plausible contact handling between objects. While PAT3D is presented on moderately sized scenes, the framework naturally extends to larger and more complex scenes. Our current results already include densely packed scenes (e.g., Fig. 12, row 2 col 2 and row 3 col 2), each involving up to 19 mutually contacting objects.
>
> A natural extension to large-scale scenes is a **top-down hierarchical optimization strategy**, where coarse-level clusters are first driven to equilibrium, followed by refinement within each cluster. We consider this a promising direction for future work.
>
> **Solver stability and runtime.**
>
> Indeed, increasing the number of objects does enlarge the contact graph and can increase solver complexity. However, the barrier-based contact simulation method we use scales effectively even in challenging scenarios. Barrier methods are specifically designed to prevent intersections during optimization, ensuring that any converged solution is by construction penetration-free. These approaches have also been validated on highly nonlinear and extreme deformation settings, demonstrating robust and reliable performance.
>
> Regarding runtime, recent advances in GPU-optimized contact simulators have significantly improved the efficiency in large-scale scenes (e.g., as shown in Fig. 5 of [1], a challenging card house scene with more than 100K mesh vertices can be simulated in real-time), and PAT3D can similarly benefit from these developments as scene sizes grow, achieving at least one order-of-magnitude faster runtime.
>
> &nbsp;
>
> *[1] Lan L, Lu Z, Yuan C, Xu W, Su H, Wang H, Jiang C, Yang Y. JGS2: Near Second-order Converging Jacobi/Gauss-Seidel for GPU Elastodynamics. arXiv preprint arXiv:2506.06494. 2025 Jun 6.*
>
> &nbsp;
>
> &nbsp;
>
>
> **Weakness 2**: Lack of quantitative study in ablation: Measurable gains introduced by each component are necessary, making audience further understand the contributions.
>
> &nbsp;
>
> Thank you for the great suggestion. We have expanded the quantitative ablation study; see Table 1 in the revision.
> Our ablation includes three settings:
>
> (1) **Raw layout** –  Raw layout directly obtained from depth prediction, which contains intersections and physically unstable configurations.
>
> (2) **Scene initialization** – Our result without simulation-in-the-loop optimization. This setting removes intersections (1) by lifting objects in an order following the scene tree. But it creates gaps between the objects and thus still lacks physical stability.
>
> (3) **Ours** – Our full pipeline by adding simulation-in-the-loop optimization on top of (2), which produces intersection-free, physically stable, and semantically consistent scenes.
>
> We also update the displacement and penetration metrics with a normalized version.
>
>
> &nbsp;
> &nbsp;
>
> Once again, thank you for your insightful review. We have incorporated the suggested changes into our paper. Please let us know if you have any additional questions or concerns. We are happy to provide further clarifications or modifications as needed.

---

### Author Response · Authors · 2025-12-03

Dear AC,

We sincerely thank you for your valuable time and efforts, especially in light of the unusual circumstances this year.

All three reviewers were positive about the manuscript after discussion.

Reviewer **etTa** acknowledged that we have addressed their concerns during the discussion period before Nov 26. Specifically, we have:

- Added an additional **metric** that reflects the physical plausibility to strengthen that our method achieves state-of-the-art performance.

- Clarified our **contributions and novelty**, as well as our advantages over additional related works like **Layout-Your-3D** and **EmbodiedGen**.

- Reported additional **algorithmic details** to address concerns regarding the simulation module.

- Expanded our discussion of **limitations and failure cases**.

Reviewer **etTa** indicated a positive stance toward our post-rebuttal manuscript.

We have also addressed the concerns raised by reviewers **D8i8** and **kPCZ**, although they did not participate in the discussion before Nov 26.

In particular, we have:

- Added a **quantitative ablation study** and clarified our method’s **scalability to large scenes** to address reviewer D8i8’s concerns.

- Clarified the algorithmic details behind the previous video results and added **two more animation videos** in the supplementary material to address reviewer kPCZ’s concerns. We also clarified the **differences and advantages** of our method against the related works like **CAST**.

Please refer to the detailed discussion in the corresponding review threads.

Regards,

Authors

---

### Meta-Review · Area_Chair_yNpE · 2026-01-07

**Summary:**

This paper presents a text-to-3D scene generation framework to produce physically plausible and simulation-ready results.
The submission received positive reviews from the reviewers.
The reviewers mainly recognize the effort to improve physical plausibility via simulation, intuitive pipeline design, and good downstream utility for robotics and simulation.
The main concerns from the reviewers were the computation overhead (D8i8, etTa), scalability to large scenes (D8i8), insufficient quantitative results (D8i8, etTa), limited visuals e.g. failure cases (etTa, kPCZ), and distinction from prior work (etTa, kPCZ).
After reading the paper, the reviewers' comments and the authors' rebuttal, the AC believes the authors' responses would have addressed the reviewers' major concerns. The reviewers have come to a consensus to recommend acceptance. The AC agrees with the evaluation.

**Reviewer Concerns:**

Reviewers' concerns mostly addressed:
- Lack of evaluation (all)
- Novelty and distinction (etTa, kPCZ)

Partially addressed concerns:
- Computation efficiency (D8i8, etTa)
- Scalability to large scenes (D8i8)

**Reviewer Scores:**

I think all reviewers would keep their original ratings (with etTa's rating raised to 6).

---

### Decision · Program_Chairs · 2026-01-26

Accept (Poster)